# An upstream secondary DNA motif within the *IL3* insulator CTCF binding site is required for enhancer-blocking insulator activity

**Sarion R. Bowers[1,2], Peter N. Cockerill** [ID][2,3]*

1 Sanger Institute, Wellcome Genome Campus, Hinxton, Cambridge, United Kingdom, 2 Leeds Institute of Molecular Research, University of Leeds, St James's University Hospital, Leeds, United Kingdom, 3 Department of Cancer and Genomic Sciences, College of Medicine and Health, University of Birmingham, Birmingham, United Kingdom

* p.n.cockerill@bham.ac.uk

## Abstract

CCCTC-binding Factor (CTCF) is the only known vertebrate protein that functions to organise the genome into distinct functional chromatin domains. A subset of CTCF sites also provide insulator activity and barrier activity to block inappropriate enhancer-promoter communication and spreading of histone modifications and non-coding transcription into loci where it is unwarranted. Paradoxically, other CTCF sites mediate enhancer and promoter communication, partly by supporting DNA loop extrusion within chromatin domains. Despite intensive study and abundant data, it remains poorly understood how CTCF directs these different functions. In this study we provide new data and mine published data that show that CTCF utilises zinc fingers 9–11 and an upstream DNA binding consensus sequence resembling CAGCTGTTCC to mediate high affinity binding and enhancer-blocking insulator activity. A single high affinity CTCF binding site from the *IL3* locus is able block *IL3* promoter activation by an upstream enhancer. Mutation of a CTGCAGCTTT sequence upstream of the CTCF core motif abolishes insulator activity. We propose that CTCF is able to confer insulator and barrier activity upon a specific subset of CTCF sites that contain the upstream DNA consensus binding motif, thereby allowing CTCF to function differently in different contexts.

## Introduction

CTCF (CCCTC-binding Factor) is a highly conserved, 11-fingered zinc finger (ZF) protein [1–3] able to use different combinations of ZFs 3–11 to bind to a highly divergent set of DNA sequences [1,4]. CTCF is a classical ZF protein that uses an array of adjacent ZFs that each bind to adjacent DNA triplets within a specific DNA consensus sequence. CTCF sites are highly diverse in that their functions range from defining genome domain boundaries, blocking spreading of chromatin modifications,

**Data availability statement:** All relevant data are within the paper and its Supporting Information files.

**Funding:** This work was supported by Biotechnology and Biological Sciences Research Council grant BB/E023002/1 awarded to P.N.C. We declare that the funders had no role in study design, data collection and analysis, decision to publish, or preparation of the manuscript.

**Competing interests:** The authors have declared that no competing interests exist.

and insulating enhancers from promoters, while conversely some CTCF sites mediate enhancer-promoter communication. CTCF plays a major role in maintaining chromosomal 3D architecture whereby it is found at the borders of independently folded chromatin domains termed Topologically Associating Domains (TADs) that are typically ~1000 kilobase (kb) in size [5–7]. These interaction domains are bordered by a subset CTCF sites that act as roadblocks to directional DNA loop extrusion through the ring of the Cohesin complex at each boundary of the TAD [8–10]. DNA loops extrude through the Cohesin ring on the NH2-terminal or ZF3 side of the bound CTCF complex which faces into the TAD. This provides a mechanism whereby loop extrusion assists enhancer interaction with genes on one side of the CTCF boundary but not the other, and allows some CTCF sites to act as enhancer-blocking insulators in functional enhancer assays [11–13].

CTCF is the principal vertebrate insulator-binding protein [14,15], and it is known to bind to most previously characterized enhancer-blocking insulators, including those in the human *IL3*/*CSF2* locus [11], the chicken beta-globin locus [16], the Igf2/H19 locus [17–20], the HLA-DQA1 locus [21], and the mouse alpha-globin locus [13]. In addition, some CTCF sites, such as ones found in the HOXA cluster and the *FOXJ3* locus, function as barrier elements segregating domains of active chromatin marked by histone H3K4me3 from the spreading of Polycomb-repressed domains of chromatin marked by H3K27me3 [22–24]. The+2.9 and +4.2 kb CTCF sites in the *IL3* insulator also block the spread of read-through non-coding transcription coming from both upstream and downstream [25]. Furthermore, CTCF is the only protein identified thus far that is directly associated with insulator function in vertebrates (other than the CTCF homolog BORIS (*CTCFL*) which is expressed only in germ cells [26]). It is through CTCF that other insulator-associated proteins, such as the Cohesin complex, are recruited to CTCF sites [11,27–29]. Despite the obvious importance of CTCF, it remains unclear how some CTCF sites can block enhancer function, while others promote enhancer function.

CTCF sites account for ~10% of all ubiquitous DNase I hypersensitive sites (DHSs) in the genome [30]. CTCF employs up to 11 ZFs to bind to its target sites, generating considerable complexity in the sequences that it can recognize. Global CTCF chromatin immunoprecipitation (ChIP) studies defined a 15 bp core sequence resembling CCACCAG(G/A)GGGGGCC that is present in all CTCF sites at least 13,000 times in the human genome [4,31,32] and this core is now known to interact with CTCF ZFs 3–7 [33–35]. However, other studies revealed that in 13–15% of cases CTCF sites make additional contacts via ZFs 9–11 to a second upstream consensus sequence resembling CAGCTGTTCC, with ZF 8 bridging a flexible length DNA linker region in between [1,11,13,34–44]. Individual deletions of ZFs 8–11 led to preferential loss of CTCF sites containing the upstream sequence [43]. The functions of the upstream CTCF motif and ZFs 9–11 remain poorly defined, but they are likely to support higher affinity binding than CTCF sites containing just the core consensus. The structure of CTCF bound to DNA is illustrated in the model in Fig 1 which is based on recently published structural data [33]. In this model CTCF snakes around the major groove of DNA with ZFs 3–7 bound to 15 bp of the core consensus while

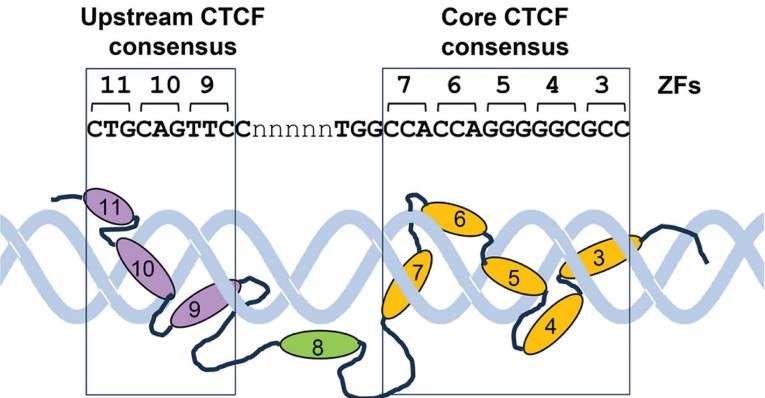

**Fig 1. A model of the proposed mechanism of binding of CTCF to DNA.** This cartoon is based on the X-Ray crystal structure of CTCF bound to DNA [33]. It portrays the approximate positions of ZFs 3–11 aligned with the most common variation of the CTCF consensus that typically has a non-specific 5 bp linker separating the 2 separate CTCF motifs. ZF domains coloured orange bound to the core consensus, and ZF domains coloured purple bound to the upstream sequence, each bind along the major groove and spiral around the DNA helix. ZF8, coloured green, lies away from the major groove and binds to phosphates across the linker region. The core consensus often has a conserved TGG-like sequence upstream of the 15 bp core but the role of this element in DNA binding is unclear as ZF7 binds to the adjacent CCA element. In practice the number of bases separating the 2 motifs can vary from 2 to 7 bp.

ZFs 9–11 bind to 9 bp of the upstream consensus. The 2 blocks of ZFs are separated by ZF8 which binds the phosphate backbone rather than the major groove.

An inspection of published studies of CTCF sites with known insulator, repressor or chromatin barrier activity reveals that they typically include the secondary upstream CTCF motif (Fig 2A) [1,11,14,16,21,37,38,40,45,46]. A recent study of CTCF sites in the alpha globin locus also found that the CTCF sites that function the most efficiently as enhancer-blocking insulators are also the CTCF sites that have the upstream motif, and this study observed that insulation is greater when the NH2-terminal or ZF3 side of the bound CTCF complex faces towards the enhancer [13]. In this orientation of the CTCF site, the enhancer is downstream of the CCACCAG(G/A)GGGGGCC sequence, and any extrusion loops are predicted to progress along the genome in the opposite direction and away from the promoter. A similar published functional screen for the insulator activity of CTCF sites inserted into the *Sox2* locus also showed a requirement for the upstream CTCF motif, but found that 2–4 tandem copies of CTCF sites were needed for insulator activity, and that the orientation was less important [12].

CTCF controls many aspects of chromatin structure, chromosomal architecture and the regulation of transcription. These functions can include regulating the intrusion of histone modifications or non-coding transcription into gene loci, and long-range chromatin looping. The complex nature of CTCF sequences and their functions makes it very difficult to dissect the individual specific functions of CTCF when sites are embedded within chromosomal DNA. Even in circular plasmids it is difficult to evaluate enhancer-blocking insulators unless they are inserted both upstream and downstream of the gene loci under study. To circumvent these difficulties, we previously developed a novel enhancer-blocking insulator assay based of the transfection of linearised luciferase reporter gene plasmids containing insulator elements in between the *IL3* promoter an upstream *CSF2* enhancer [11]. Using this model we previously reported that insertion of 3 tandem copies of the *IL3*+2.9 kb CTCF site completely blocked *CSF2* enhancer activity in linear plasmids, and blocked 80% of activity in circular plasmids [11]. This model system was able to assay for the enhancer-blocking activity of CTCF in isolation from other activities relating to chromatin modifications or chromatin architecture.

In this new study we employed one of the enhancer-blocking CTCF sites from the *IL3* insulator [11] as a model to investigate the functions of the secondary upstream CTCF binding motif. We first showed that *in vitro* CTCF binding is greatly

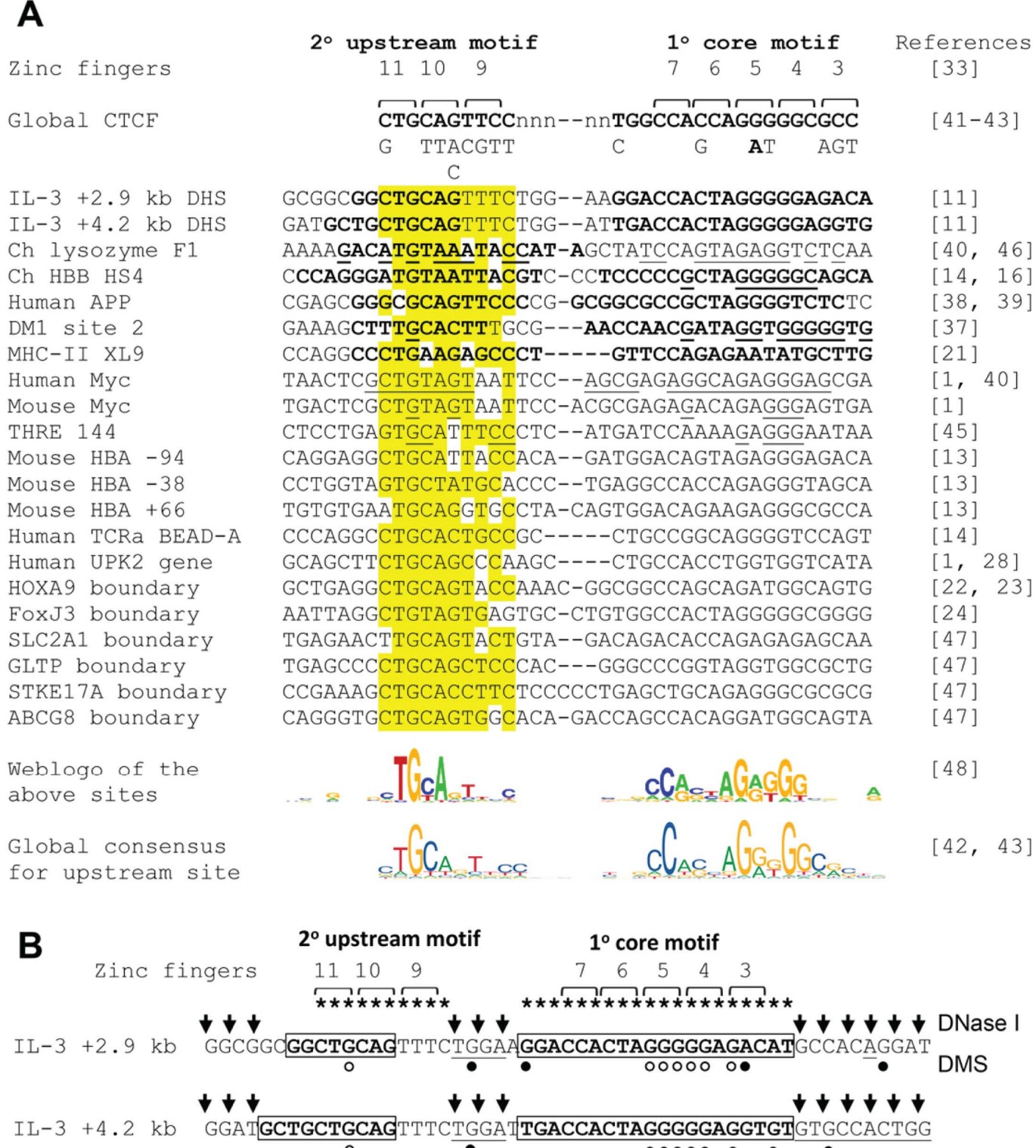

**Fig 2. Conservation among insulator and boundary CTCF sites.** (A) Comparison of sequence homology between previously characterized CTCF binding sites defined as insulators, repressors or chromatin boundaries [1,11,13,14,16,21,24,28,37,40,45,47]. The three lines of sequence in the global CTCF consensus above the table represent the predominant alternative bases allowed at each position with the preferred bases shown above

in uppercase [41–43]. Regions protected in published DNase I or DMS footprinting assays are highlighted in bold. Underlined bases represent bases where methylation or carboxyethylation blocks binding of CTCF in interference assays. Above the table are the predicted binding sites of Zn finger domains 3–7 and 9–11 (indicated by brackets and numbers) [33]. Zn Finger 8 is not shown because this domain sits away from the DNA bases above the variable length spacer region. Zn fingers 3–7 bind the core CTCF consensus sequence common to all CTCF sites (primary motif) and Zn fingers 9–11 bind the secondary upstream consensus sequence which is present in ~13% of CTCF sites. Underneath the table are shown a web logo [48] derived from the sequences in the table, and the published global consensus sequence for CTCF sites that contain the upstream binding site [42,43]. (B) Locations of the core and upstream CTCF motifs within CTCF sites found within the human IL-3 gene insulator [11]. The boxes represent regions protected from DNase I in footprinting assays. G bases protected from modification by DMS in *in vivo* footprinting assays are indicated by open circles. Regions hypersensitive to DNase I (arrows) or modification by DMS (filled circles) are also indicated.

reduced in the absence of this upstream DNA motif. We next employed our previously described enhancer-blocking insulator assay based on linearised DNA fragments containing the *CSF2* enhancer and the *IL3* promoter [11]. We demonstrated that a single CTCF site from the *IL3* insulator was sufficient to confer robust enhancer-blocking insulator activity and that the secondary upstream consensus sequence was essential for this activity. This observation now further builds upon our understanding of mechanisms that determine why only a subset of CTCF sites function as insulators.

## Results

### The upstream CTCF motif is a common feature of enhancer insulators and chromatin barriers

The role of the upstream CTCF consensus sequence at insulators remains poorly understood. To find published evidence for its significance we performed a brief survey of well-defined CTCF-dependent insulators and found that they typically included a near ideal match to the 10 bp upstream binding site (highlighted in yellow in Fig 2A). Fig 2A also summarises previously published *in vitro* DNA binding data and *in vivo* footprinting data confirming contacts between CTCF and the upstream motif where ZFs 9–11 are predicted to bind. Regions protected by CTCF from either DNase I digestion and/or methylation by DMS *in vivo* are highlighted in bold, whereas regions where CTCF binding is blocked by *in vitro* modification of DNA by either methylation or carboxyethylation are underlined. For the two *IL3* insulator CTCF sites a more detailed summary of these chromatin structure analyses is depicted in Fig 2B, and the evolutionary conservation of these sites is depicted in Table S3 in S1 File. The *IL3* insulator encompasses 2 nearly identical CTCF binding sites that include both the core consensus and an upstream CTGCAGTTTC motif (Fig 2A and B). Each of these motifs is highly conserved across mammalian species for both sites (S3 Table) in S1 File. Our previous study of this insulator employed *in vivo* footprinting using dimethyl sulphate to confirm that CTCF bound to both the upstream and core motifs, while the flanking sequences were hypersensitive to DNaseI (summarised in Fig 2B) [11]. Most other studies of nuclease accessibility at insulator CTCF sites similarly find that the regions occupied by ZF 3–7 and ZF 9–11 are separated by an accessible variable length linker approximately 8 bp in length where ZF8 is thought to sit away from the major groove of the linker DNA [11,13,33,44].

Fig 2A also lists CTCF sites within previously defined chromatin boundary elements from the *HOXA9* and *FOXJ3* loci that similarly include the upstream CTCF motif [22–24]. To find further evidence for the presence of the upstream CTCF motif at histone H3 K27me3 chromatin boundaries we next performed a limited scan of the publicly available genome data from a global study of histone methylation and CTCF binding, provided by the authors online via a link to the UCSC genome browser (https://dir.nhlbi.nih.gov/papers/lmi/epigenomes/hgtcell.aspx) [24,47]. This allowed us to identify several additional CTCF sites with conserved upstream CTCF motifs at H3K27me3 chromatin boundaries (Fig 1 and S1 Figure in S1 File). We examined the conservation of a sample of the above sites and found that like the *IL3* insulator (Fig 2A and S3 Table in S1 File), the upstream CTCF site is highly conserved in at least the UPK2, HOXA9, FOXJ3 and ABCG8 CTCF sites (S4 to S6 Tables in S1 File). The sequence logo shown underneath Fig 2A is derived from all of the insulator sites listed in Fig 2A. This motif is essentially identical to that derived from global analyses of the upstream consensus, suggesting a strong functional association.

During the revision of this manuscript, another genome wide study of CTCF sites has been published identifying an additional subset of CTCF sites located at heterochromatin boundaries separating active genes from heterochromatin marked by histone H3K9me2 [49]. Although this study did not specifically examine the role of the upstream CTCF motif, it did present genomic data for 2 examples of CTCF sites in this category of barrier element at the human *BICD1* and *FMC1* loci. Our subsequent DNA sequence analyses of these sites show that they also have near complete copies of the upstream CTCF site (Fig S7 in S1 File). Hence, CTCF sites carrying the upstream motif are highly likely to function as barriers to both of the two major classes of repressed chromatin found in vertebrates.

### The upstream CTCF motif promotes high affinity binding to DNA

Using the *IL3* insulator as a model, we employed Electrophoretic Mobility Shift Assays (EMSAs) with Jurkat human T cell nuclear extracts to investigate the relative contributions of the primary core (1°) and secondary upstream (2°) motifs to *in vitro* CTCF binding. Because CTCF is a ZnF protein, EMSAs were performed in the presence of $Zn^{2+}$, which we previously found to be essential for efficient *in vitro* binding of CTCF using nuclear extracts [11]. We synthesised various mutated derivatives of the +2.9 kb *IL3* CTCF site for use in EMSAs and included intact and mutated versions of the chicken beta-globin HS4 CTCF site as controls (Fig 3A). The identity of the CTCF complex was confirmed using CTCF antibodies and control IgG (Fig 3B). EMSAs demonstrated that *in vitro* CTCF binding was highly dependent on the presence of the conserved core motif, and binding was abolished if the core consensus was mutated in either the *IL3* or the HS4 CTCF site (Δ1° and ΔHS4) (Fig 3B). High affinity CTCF binding was dependent upon maintaining the upstream motif intact, whereby binding was reduced 3-fold when it was mutated (Δ2°) (Fig 3B). These binding data were quantified by densitometry (Fig 3C). The relative strength of binding of the intact and mutated CTCF sites in EMSAs was also evaluated by including 3, 10 or 30 ng of unlabelled oligonucleotides as competitors (Figs 2D - F). The mutation of the core sequence eliminated all competing binding activity while mutation of the upstream sequence resulted in a modest decrease in competition.

The unmanipulated EMSA images are shown in S2 Fig in S1 File.

### The secondary upstream CTCF motif is essential for enhancer-blocking *IL3* insulator function

We next dissected the relative contributions that the upstream and core consensus sequences made towards the enhancer-blocking activity of the *IL3* +2.9 kb insulator CTCF site. We performed transfection assays in Jurkat T cells with plasmids containing a 717 bp fragment of the inducible human *CSF2* enhancer (GME) upstream of the −559 to +50 region of the inducible human *IL3* promoter in front of the luciferase gene in pXPG (Fig 4). The activity of the inducible *IL3* promoter was measured in linearised plasmids 40 hours after transfection into Jurkat T cells, following stimulation for 8 h with 20 ng/ml PMA and 1 mM calcium ionophore A23187 to induce T cell receptor signalling. Enhancer-blocking insulation activity was measured after placing a single CTCF site in the reverse orientation in between the enhancer and promoter. In this configuration the CTCF ZF 3–7 binding sequence CCACTAGGGGGAGACA is facing towards the enhancer, which is predicted to maximise the insulation activity of CTCF. Insertion of just a single copy of the *IL3* CTCF site was sufficient to almost entirely block *CSF2* enhancer activity (Fig 4). However, mutation of either the upstream motif or the core CTCF consensus sequence abolished enhancer-blocking insulator activity (Fig 4). The activities of these constructs were not measured in non-stimulated cells because the *IL3* promoter and *CSF2* enhancer are essentially inactive in the absence of stimulation [50,51].

## Discussion

The aim of this study was to further raise the consciousness of the scientific community about the potential role of the upstream CTCF consensus in insulator and barrier function. Despite numerous studies over the last 25 years revealing binding of CTCF to an upstream CTCF motif at known insulators, the role and mechanism of action of this secondary

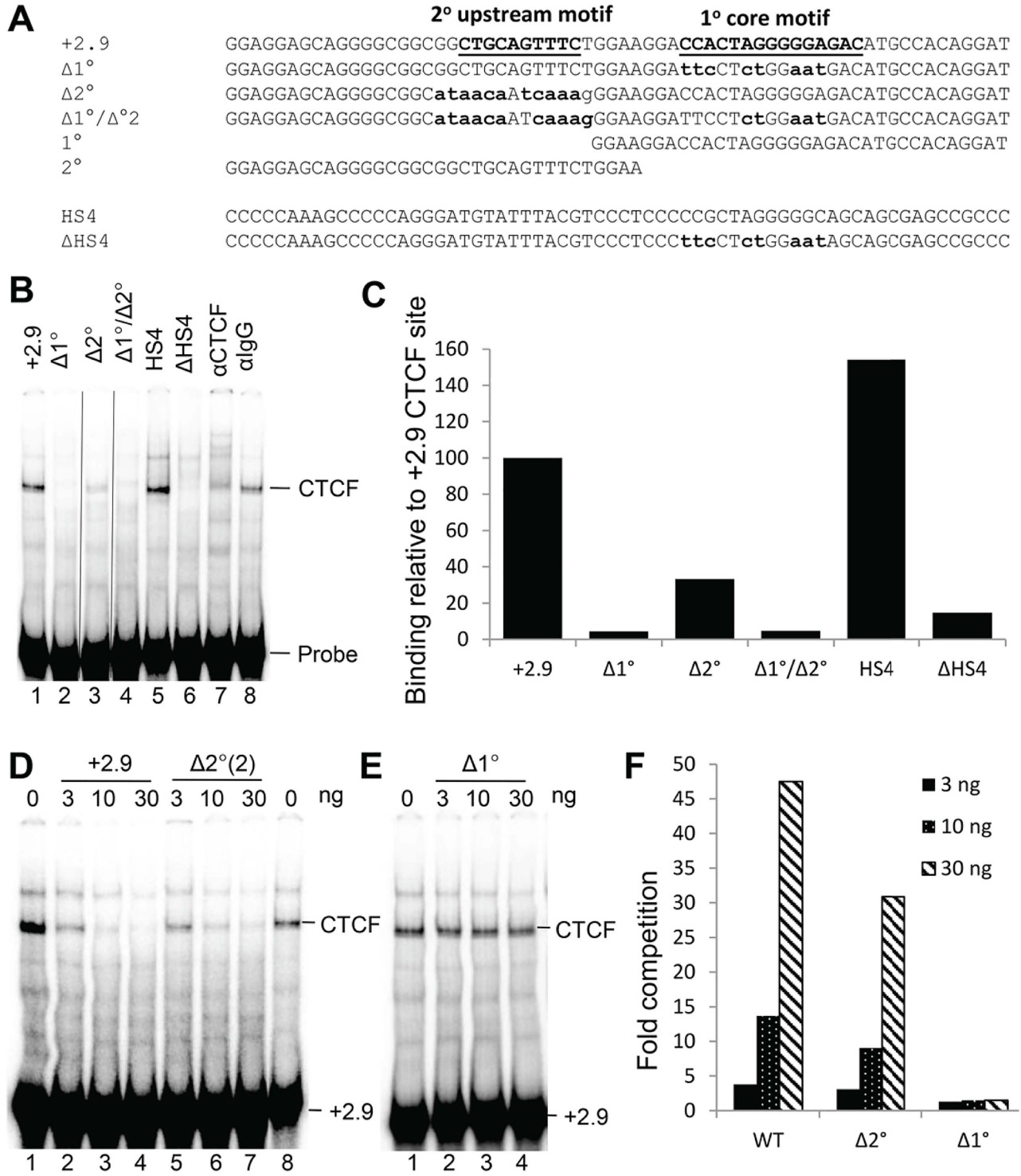

**Fig 3. EMSAs of CTCF binding to the *IL3* insulator CTCF site.** (A) Sequences of CTCF EMSA probes used in this study. Mutated residues are shown in bold lowercase and the two consensus sequences are underlined. (B and C) EMSAs using intact or mutated *IL3*+2.9 kb CTCF sites. EMSAs included 4 μg of Jurkat nuclear extract with either the +2.9, Δ1°, Δ2°(1), Δ2°(2), Δ1°/Δ2°(2), HS4, or ΔHS4 probes (B). Specificity of CTCF binding was confirmed by incubating 4 μg of nuclear extract with 2 μl of anti-CTCF or control IgG antibodies, and by comparison to the chicken beta globin HS4 CTCF site (B). The vertical black lines indicated parts of the gel where 2 irrelevant lanes were deleted. Binding data was quantified by densitometry (C). (D-F)

Competition assays of CTCF binding. EMSAs included 4 µg of Jurkat nuclear extract in the presence of increasing amounts unlabelled DNA competitor as indicated above. (F) Densitometric quantification of the EMSAs in panels D and E depicting the efficiency of the mutated +2.9 CTCF sites to compete with wild-type +2.9 site. Values are expressed as the fold decrease in the presence of competitor.

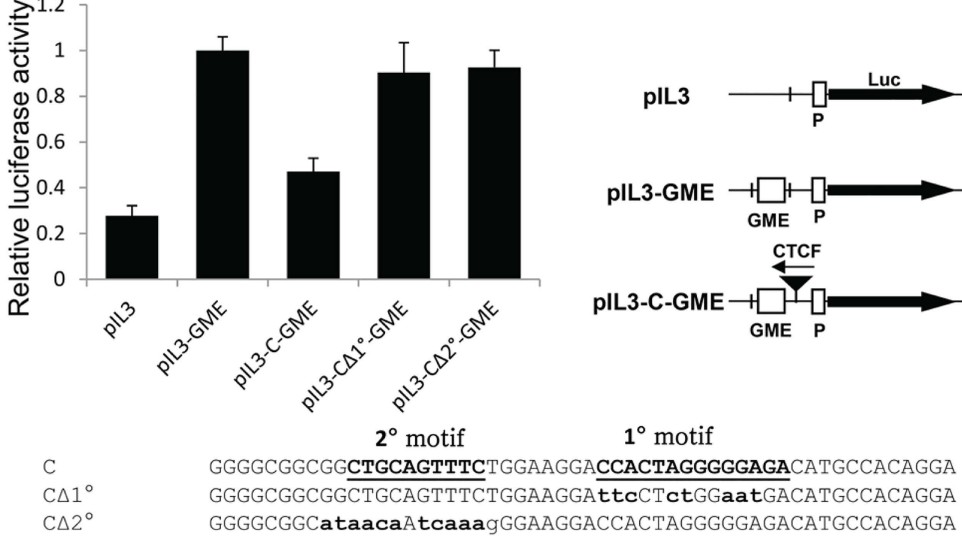

**2° motif** **1° motif**

```
C      GGGGCGGCGGCTGCAGTTTCTGGAAGGACCACTAGGGGGAGACATGCCACAGGA
CΔ1°   GGGGCGGCGGCTGCAGTTTCTGGAAGGAttcCTctGGaatGACATGCCACAGGA
CΔ2°   GGGGCGGCataacaAtcaaagGGAAGGACCACTAGGGGGAGACATGCCACAGGA
```

**Fig 4. Enhancer blocking assay of a CTCF site.** Linearised luciferase reporter gene plasmids containing the human IL-3 gene promoter (pIL3) were assayed in Jurkat T cells. As depicted in the cartoon at the right, derivatives of pIL3H contained either just the *CSF2* enhancer (pIL3-GME) or the enhancer plus a CTCF site inserted between the promoter and the enhancer (pIL3-C-GME). CTCF site plasmids contained either the intact *IL3*+2.9 kb CTCF site, or this CTCF with either the primary or the secondary consensus motif mutated, as indicated below. The CTCF sites are inserted into the plasmids in the opposite orientation to that shown which means that the NH2 terminus of bound CTCF faces towards the enhancer. Luciferase activities were measured 40 h after transfection plus an additional 8 h after stimulating transfected Jurkat cells with PMA and calcium ionophore. Luciferase activities were expressed relative to pIL3-GME having a value of 1, and represent the mean of 6 transfections, using two independently derived DNA clones. Error bars indicate Standard Deviation. Data was normalised by comparison with the cotransfected Renilla luciferase control plasmid pRL-TK.

binding site is still not fully understood or appreciated. Our findings are in agreement with another recent study of CTCF sites in the mouse alpha globin locus which similarly found a correlation between the presence of the upstream CTCF motif and the ability of CTCF sites to function as enhancer-blocking insulators when inserted into the genome [13]. They also agree with the findings of a genomic screen for insulators inserted into the mouse *Sox2* locus whereby CTCF sites that functioned as insulators, or were found at TAD boundaries, were also those CTCF sites that included the upstream CAGCTGTTCC-like motif [12]. However, one major difference to our study was that the *Sox2* study found that single CTCF sites had no enhancer-blocking insulator activity and four tandem CTCF sites were required to block 38% of gene expression [12]. In our study, a single CTCF site functioned as an insulator to block 74% of enhancer activity in the context of non-integrated linear DNA, although we also previously observed that 100% insulation was only achieved when 3 tandem copies were used [11]. The above study of CTCF sites in the alpha globin locus also found that insulation strength increased with insertion of multiple CTCF sites [13].

## The mechanism of insulation by CTCF remains an enigma

The mechanism of action of insulation mediated via the upstream CTCF motif is far from clear. In our study we found that the upstream motif was needed for high affinity binding, but the work of others indicated that high affinity binding alone is not sufficient for insulator activity [12,13]. This raises the possibility that CTCF has 2 independent functional

DNA binding domains that contribute different activities. This is supported by the fact that CTCF uses ZFs 3–7 and ZFs 9–11 as two independent clusters of DNA binding domains whereby the ZFs bind side by side to adjacent 3 bp ZF recognition sequences to the two separate motifs which are interrupted by a variable length (~7–10 bp) DNA spacer bridged by the more flexible ZF8 which interacts with phosphates rather than the bases of the major groove. The data suggest that interactions with the core CTCF motif via ZFs 3–7 are essential for all CTCF binding whereas binding via ZFs 9–11 contributes additional functionality at a subset of sites. It is evident that the 2 DNA binding domains bind somewhat independently because the DNA linker is of flexible length and is highly accessible to nucleases when CTCF is bound. The fact that many CTCF sites also have a TGG or CGG-like adjacent to the CCA bound by ZF7 [4] hints at additional possible modes of binding by CTCF that might involve yet another mode of binding involving ZF8 (Figs 1 and 2A).

CTCF sites that function as insulators or TAD boundaries normally function in a specific orientation whereby the core motif and ZF3 face into the loop [8]. This may be related to the fact that CTCF recruits cohesin, and DNA loop extrusion through the cohesin ring proceeds by drawing in DNA from the ZF3 end of the CTCF/cohesin complex [8,13], which is downstream and not upstream of the CTCF core motif [10]. As TAD DNA loop extrusion can extend for hundreds of kilobases, most CTCF sites probably have the ability to pass through the cohesin ring. Taken together, these observations support a model whereby high affinity binding of ZFs 9–11 to the upstream motif provides a double lock to the CTCF complex, stabilising its binding, and rendering domain boundaries and loop ends more stable. Enhancer-blocking insulation also works most effectively when ZF3 is facing towards the enhancer [13]. However, it seems unlikely that enhancer insulation can be regulated solely on the basis of effects on loop extrusion and enhancer looping to promoters because many insulator assays have the enhancer less than 1 kb from the promoter. Nevertheless, cohesin almost certainly contributes to insulator activity, and co-association with the cohesin protein Rad21 provides a better predictor of insulator activity that just strength of CTCF binding [13].

There is also evidence that CTCF binds less rigidly to CTCF sites lacking the upstream motif [52]. Some CTCF sites bind in a highly tissue-specific manner that relies on cooperation with other transcription factors binding to adjacent sequences. A study of these dynamic CTCF sites in blood cells found that only 6% of these sites included the upstream motif whereas most insulator and barrier CTCF sites have it, compared to 13–15% of CTCF sites globally that include the upstream motif [52].

While we still have a poor understanding of the molecular mechanisms of enhancer blocking, there may also be a role for CTCF ZF8 which interacts with the phosphate backbone above the minor groove of the linker DNA separating the core and upstream motifs [33]. The other ZFs interact with the DNA bases in the major groove. It is likely that any structure formed by ZF8 on DNA will itself be much more rigid, and fixed like a bridge above the linker, when both motifs are tightly bound. Such a rigid structure would be expected to behave differently than one where binding via ZFs 8–11 is dynamic.

## The upstream CTCF binding motif is not conserved in insects

The role of the upstream CTCF binding motif at insulators described here may be restricted to vertebrates. A study of CTCF ChIP peaks in Drosphila S2 cells found no enrichment for the closely linked CTGCAGTTCC-like motifs identified in CTCF sites in human cells, but just the same core consensus motif as is found in vertebrates [53]. This is most likely due to the fact that ZFs 9–11 are not as highly conserved between human and Drosophila CTCF as are ZFs 3–7 [54]. This includes 2 amino acids in ZF9 and 1 amino acid in ZF11 that are predicted to contact DNA in human CTCF [53]. Although Drosophila do express both CTCF and cohesin, they may not interact or function in the same way as they do in vertebrates. This is partly because Drosophila utilise multiple insulator proteins and not just CTCF [53,55]. Drosophila insulator CTCF sites also showed a partial dependency upon additional unidentified flanking sequences located within 90 bp of the core motif [53].

## CTCF is a multifunctional protein

Taken together, our data and recent published studies all predict that CTCF can use two distinct modes of binding to contribute very different and seemingly contradictory functions. Binding via just ZFs 3–7 may support more dynamic binding, and generate a structure able to pass through cohesin rings. This mode of binding is more likely to fit with the vast majority of CTCF sites that probably assist enhancer-promoter communication. Conversely, binding that also includes tight binding to ZFs 9–11 may generate a more rigid structure that is either unable to pass though cohesin rings or is tightly bound to cohesin itself. This mode of binding appears to support insulator or barrier function. Future studies will need to determine the structures that CTCF adopts when bound to DNA together with its partners to gain a better understanding of the mechanisms of insulation.

## Conclusions

Our new data and analyses of previously published data collectively provide strong evidence that the upstream CTCF consensus present in ~13–15% of all CTCF sites plays a major role in both enhancer insulator and chromatin barrier function.

## Materials and methods

### Electrophoretic Mobility Shift Assays (EMSAs)

Double stranded oligonucleotides containing the wild type sequence, mutations or truncations of the +2.9 kb CTCF binding site from the IL-3/GM-CSF insulator [11] or chicken β-globin gene [14] were labeled by end-filling with α-[32P]-dCTP plus unlabelled dATP, dGTP and dTTP. The oligonucleotide sequences were as follows:
+2.9: TCGATATCCTGTGGCATGTCTCCCCCTAGTGGTCCTTCCAGAAACTGCAGCCGCCGCCCCTGCTCCTCCCTCGA
Δ2°(1): TCGATATCCTGTGGCATGTCTCCCCCTAGTGGTCCTTCCAGAAAATTGTTATGCCGCCCCTGCTCCTCCCTCGA
Δ2°(2):
TCGATATCCTGTGGCATGTCTCCCCCTAGTGGTCCTTCCCTTTGATTGTTATGCCGCCCCTGCTCCTCCCTCGA
Δ1°: TCGATATCCTGTGGCATGTCATTCCAGAGGAATCCTTCCAGAAACTGCAGCCGCCGCCCCTGCTCCTCCCTCGA
HS4: TCGATCCCCCAAAGCCCCCAGGGATGTATTTACGTCCCTCCCCCGCTAGGGGGCAGCAGCGAGCCGC-
CCCTCGA
ΔHS4:
TCGATCCCCCAAAGCCCCCAGGGATGTATTTACGTCCCTCCCTTCCTCTGGAATAGCAGCGAGCCGCCCCTCGA
Δ1°/Δ2°: TCGATATCCTGTGGCATGTCATTCCAGAGGAATCCTTCCCTTTGATTGTTATGCCGCCCCTGCTCCTC-
CCTCGA.

Nuclear extract from unstimulated Jurkat cells were prepared as previously described [56]. 4 μg of nuclear extract was incubated with 4 μg of poly(dI-dC) and 100 ng of an unrelated 22 bp oligonucleotide duplex competitor (containing the GM450 Runx-1 element from the GM-CSF enhancer [57] for 10 minutes before addition of 0.2 ng of labeled oligonucleotide in a 17 μl reaction mixture containing 10% glycerol, 20 mM HEPES, 30 mM KCl, 30 mM NaCl, 0.1 mM ZnSO4, 0.1 mM MgCl2, 1% dithiothreitol, 0.1 mM phenylmethylsulfonyl fluoride, 5 μg/ml aprotinin, 5 μg/ml leupeptin for 15 minutes at room temperature (~ 22°C). 4% polyacrylamide gels containing 25 mM Tris borate, 0.5 mM EDTA, were pre-electrophoresed at 10 V/cm for 1 h and run at 10 V/cm for 1 h 20 min at room temperature. Gels were fixed in 0.1% cetyl trimethylammonium bromide, 50 mM sodium acetate before drying and visualization on a phosphorimager screen. Supershift assays were carried out by addition of 2 μl of CTCF antibody (07–729 Millipore) or control IgG- (12–370 Millipore) to 4 μg of Jurkat nuclear extract and incubation for 10 min at room temperature before subsequent addition of poly(dI-dC) and oligonucleotide. Competition assays were carried out by addition of the specified amount of unlabelled oligonucleotide competitor in 1 μl to 4 μg of nuclear extract and incubated for 15 mins at room temperature before addition of 0.2 ng of +2.9 labeled oligonucleotide.

Relative binding and competition efficiency were determined by densitometric analysis using Quantity One (Bio-Rad) software.

## Enhancer-blocking assays

The insulator function of individual CTCF sites was tested using linearised firefly luciferase reporter gene plasmids as previously described [11]. Briefly, gene fragments cloned into pXPG [58] were cotransfected with the Renilla luciferase control plasmid pRL-TK into Jurkat cells which were then cultured for 40 h before stimulation for 8 h with 20 ng/ml Phorbol 12-myristate 13-acetate (PMA) and 1 mM calcium ionophore A23187. Cell extracts were then harvested for dual luciferase activity. The plasmid pIL3 (also known as pIL3H [51]) contains the −559 to +50 region of the human IL-3 gene promoter in the Hind III site of pXPG [58], and pIL3-GME contains the 717 bp Bgl II fragment of the human GM-CSF gene enhancer cloned upstream in the Bgl II site of pIL3. The pIL3-C-GME series of insulator plasmids were made by inserting a single copy of the indicated DNA segments containing the intact or disrupted IL-3 + 2.9 kb CTCF site into the Xho I site of pIL3-GME using Xho I adapters at each end. The CTCF sites are inserted into the plasmids in the opposite orientation to that shown in Fig 1A and Fig 2 which means that the NH2 terminus of bound CTCF and ZF3 face towards the enhancer while ZF11 faces towards the promoter. Firefly luciferase plasmids were linearised with Aat II prior to transfection. Relative luciferase activity was calculated as the ratio of firefly and Renilla luciferase activity.

## Supporting information

**S1 File. Supporting information Bowers and Cockerill.**
(PDF)

## Author contributions

**Conceptualization:** Peter N. Cockerill.

**Data curation:** Peter N. Cockerill, Sarion R. Bowers.

**Formal analysis:** Peter N. Cockerill, Sarion R. Bowers.

**Funding acquisition:** Peter N. Cockerill.

**Investigation:** Peter N. Cockerill, Sarion R. Bowers.

**Methodology:** Peter N. Cockerill.

**Project administration:** Peter N. Cockerill.

**Supervision:** Peter N. Cockerill.

**Visualization:** Peter N. Cockerill.

**Writing – original draft:** Peter N. Cockerill.

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
