## [Decision Letter · Decision Letter 0]

8 Jan 2026

Dear Dr. Cockerill,

Thank you for submitting your manuscript to PLOS ONE. After careful consideration, we feel that it has merit but does not fully meet PLOS ONE’s publication criteria as it currently stands. Therefore, we invite you to submit a revised version of the manuscript that addresses the points raised during the review process.

The data presented in this manuscript are compelling and constitute a meaningful contribution to the existing body of work in this area. Overall, the study is well executed and clearly written. With a small number of minor revisions and the addition of one clarifying experimental component, the manuscript would be strengthened further.

We look forward to receiving your revised manuscript.

Kind regards,

Purnima Singh, PhD

Academic Editor

PLOS One

Journal Requirements:

Biotechnology and Biological Sciences Research Council grant BB/E023002/1 awarded to P.N.C.

This work was supported by Biotechnology and Biological Sciences Research Council grant BB/E023002/1 awarded to P.N.C.

Biotechnology and Biological Sciences Research Council grant BB/E023002/1 awarded to P.N.C.

Reviewers' comments:

Reviewer's Responses to Questions

**Comments to the Author**

1. Is the manuscript technically sound, and do the data support the conclusions?

Reviewer #1: Yes

Reviewer #2: Yes

2. Has the statistical analysis been performed appropriately and rigorously?

Reviewer #1: N/A

Reviewer #2: Yes

3. Have the authors made all data underlying the findings in their manuscript fully available?

Reviewer #1: Yes

Reviewer #2: Yes

4. Is the manuscript presented in an intelligible fashion and written in standard English?

Reviewer #1: Yes

Reviewer #2: Yes

Reviewer #1: CTCF binding at DNA sequences not only insulates enhancer-promoter interactions but also plays essential role in chromatin boundary formation. It is not known whether or how those functions are distinguished at the molecular level and if any DNA sequence can specify TAD function over insulator function. Drs. Bowers and Cockerill revisit the IL3 insulator they characterized earlier. A secondary DNA sequence upstream of the CTCF core motif binding site was recently reported to be part of the strongest enhancer–promoter blocker CTCF binding sites. Using EMSA the authors find that this sequence in the IL3 insulator is, indeed, important for CTCF binding. Based on similarity they predict that CTCF ZF9-11 are in contact with this secondary motif (in addition to ZF4-7 at the core motif). In luciferase assays they find that the secondary motif is important for insulating enhancer-promoter interaction in a linear transfected DNA template. They also provide genome browser views from a published ChIP-seq dataset suggesting that such composite CTCF sites are not only present in known insulators but are also located at several known chromatin boundaries, suggesting a function at TAD formation. The authors then provide an interesting discussion based on their findings and observations. This Discussion should motivate follow-up studies that may include a more in-depth bioinformatics analysis of the two types of CTCF binding sites, whether the composite site always occurs at boundaries, whether it co-occurs with cohesin, and whether the simple site (without secondary motif) forms boundaries. In addition, future studies may test the role of the secondary sequence in boundary function by genetics.

Reviewer #2: Summary

In this study, the authors present data on the molecular role of an upstream secondary DNA binding motif for the CTCF insulator protein in vertebrates. Using a combination of bioinformatics, in vitro binding assays and an in vivo cell-based reporter system, they demonstrate a clear role for the secondary binding motif in mediating high affinity CTCF binding and functional activity as an enhancer-blocking insulator at IL3. The methods and results are scientifically sound, detailed, clearly presented and will be of interest to the transcriptional regulation, genome organization and developmental biology communities. Minor comments and one experimental suggestion are provided below to enhance the paper.

Minor comments

1. Adding a more graphical representation to summarize the CTCF protein domain organization and previously characterized binding modes (of the ZFs) would enhance Figure 1.

2. The sub-section “The upstream CTCF motif is a common feature of enhancer insulators and chromatin barriers” should be moved to the beginning of the Results section.

3. A control should be added to the luciferase assay experiments described in lines 182-188. The CTCF site should also be tested in the forward orientation to complement the reverse orientation results. Based on the hypothesis presented, this should reduce the insulator activity and importantly would shed light on the functional role of the molecular organization of the protein-DNA complex.

**Do you want your identity to be public for this peer review?** For information about this choice, including consent withdrawal, please see our Privacy Policy

Reviewer #1: No

Reviewer #2: No

---

## [Author Response · Author response to Decision Letter 1]

15 Jan 2026

RESPONSES TO THE REVIEWERS

Reviewer #1:

We thank reviewer 1 for their supportive comments.

There were no modifications requested by reviewer 1.

Reviewer #2: Summary

We also reviewer 2 for their supportive comments which are summarised here:

The methods and results are scientifically sound, detailed, clearly presented and will be of interest to the transcriptional regulation, genome organization and developmental biology communities.

We note that there were no substantial revisions requested, but just some minor comments.

As outlined below we have either (a) modified the paper according to the requests, (b) been unable to add new data, or (c) explained why additional data is not required.

Minor comments and one experimental suggestion are provided below to enhance the paper.

Minor comments

1. Adding a more graphical representation to summarize the CTCF protein domain organization and previously characterized binding modes (of the ZFs) would enhance Figure 1.

This was a very good suggestion. We have added a new figure 1 where we have drawn a model of CTCF binding to DNA based closely on the actual structural data from Yang et al. This appears in lines 80-95.

2. The sub-section “The upstream CTCF motif is a common feature of enhancer insulators and chromatin barriers” should be moved to the beginning of the Results section.

This was also a good suggestion. We have moved the description of known insulators to the beginning of the results. This required some juggling of text but the paper reads better now.

While doing this we also discovered that a new publication by Shin et al on CTCF sites chromatin barriers had been published in 2026 after we submitted this manuscript. This describes an additional major class of barrier that blocks spread of H3K9me2 which now adds to our previous description of barriers that block spread of H3K27me3. The new text appears in lines 188-196.

3. A control should be added to the luciferase assay experiments described in lines 182-188.

We disagree with the reviewer because the only control required is the pIL3 construct that contains the promoter alone. All the constructs are assayeded under the same conditions, so the data is already internally controlled. Each assay is also normalised to a Renilla luciferase cotransfected control. These constructs are inactive in the absence of stimulation, so there is no point in show nil values for unstimulated cells. Nevertheless, we added an extra sentence at lines 246-248 to the manuscript explaining that these constructs are inactive in the absence of stimulation and we cited out studies proving this point. There is no point repeating our published data.

The CTCF site should also be tested in the forward orientation to complement the reverse orientation results. Based on the hypothesis presented, this should reduce the insulator activity and importantly would shed light on the functional role of the molecular organization of the protein-DNA complex.

This is also a great suggestion but we do not have the resources to make additional constructs or perform new assays. This is beyond the scope of this study which aimed at suggesting future studies that other groups could perform. This was listed as a minor suggestion so we assume this was optional.

RESPONSES TO THE EDITOR

The editor suggested that we include a lab protocol for the studies described in this manuscript. However, we consider this to be not essential because all of the methods have been previously published by us and represent standard protocols used by most molecular biology labs. Furthermore, we do not have a single lab protocol but many linked protocols, and the effort to do this would be equal to writing a new manuscript.

This study was funded by the UK BBSRC.

Can the editor please include a funding statement in the appropriate place that reads:

This work was supported by Biotechnology and Biological Sciences Research Council grant BB/E023002/1 awarded to P.N.C. We declare that the funders had no role in study design, data collection and analysis, decision to publish, or preparation of the manuscript.

We note that the submission guidelines do not ask for a section called “Funding statement but just acknowledgements. The funding section of the submission form asked for funders but not the role of the funders as far as we could see.

In the acknowledgements we already state that.

This work was supported by Biotechnology and Biological Sciences Research Council grant BB/E023002/1 awarded to P.N.C.

If this is not appropriate then remove the acknowledgements section.

---

## [Decision Letter · Decision Letter 1]

20 Jan 2026

An upstream secondary DNA motif within the IL3 insulator CTCF binding site is required for enhancer-blocking insulator activity

PONE-D-25-67228R1

Dear Dr. Cockerill,

We’re pleased to inform you that your manuscript has been judged scientifically suitable for publication and will be formally accepted for publication once it meets all outstanding technical requirements.

Kind regards,

Purnima Singh, PhD

Academic Editor

PLOS One

Additional Editor Comments (optional):

Reviewers' comments:

Reviewer's Responses to Questions

**Comments to the Author**

Reviewer #2: All comments have been addressed

2. Is the manuscript technically sound, and do the data support the conclusions?

Reviewer #2: Yes

3. Has the statistical analysis been performed appropriately and rigorously?

Reviewer #2: Yes

4. Have the authors made all data underlying the findings in their manuscript fully available?

Reviewer #2: Yes

5. Is the manuscript presented in an intelligible fashion and written in standard English?

Reviewer #2: Yes

Reviewer #2: The authors have addressed all prior (minor) concerns satisfactorily. I still believe the opposite orientation control suggested would provide additional mechanistic insight, but acknowledge that it may lie beyond the scope of the current manuscript.

**Do you want your identity to be public for this peer review?** For information about this choice, including consent withdrawal, please see our Privacy Policy

Reviewer #2: No

---

## [Editor Report · Acceptance letter]

PONE-D-25-67228R1

PLOS One

Dear Dr. Cockerill,

I'm pleased to inform you that your manuscript has been deemed suitable for publication in PLOS One. Congratulations! Your manuscript is now being handed over to our production team.

Kind regards,

on behalf of

Dr. Purnima Singh

Academic Editor

PLOS One